# Using Python Modules in Real-Time Plasma Systems for Fusion

**DOI:** 10.3390/s22186847

**Published:** 2022-09-10

**Authors:** Nicolo Ferron, Gabriele Manduchi

**Affiliations:** 1Centro Ricerche Fusione, Universita di Padova, 35127 Padova, Italy; 2Consorzio RFX, 35127 Padova, Italy

**Keywords:** real-time systems, Python

## Abstract

One of the most important applications of sensors is feedback control, in which an algorithm is applied to data that are collected from sensors in order to drive system actuators and achieve the desired outputs of the target plant. One of the most challenging applications of this control is represented by magnetic confinement fusion, in which real-time systems are responsible for the confinement of plasma at a temperature of several million degrees within a toroidal container by means of strong electromagnetic fields. Due to the fast dynamics of the underlying physical phenomena, data that are collected from electromagnetic sensors must be processed in real time. In most applications, real-time systems are implemented in C++; however, Python applications are now becoming more and more widespread, which has raised potential interest in their applicability in real-time systems. In this study, a framework was set up to assess the applicability of Python in real-time systems. For this purpose, a reference operating system configuration was chosen, which was optimized for real time, together with a reference framework for real-time data management. Within this framework, the performance of modules that computed PID control and FFT transforms was compared for C++ and Python implementations, respectively. Despite the initial concerns about Python applicability in real-time systems, it was found that the worst-case execution time (WCET) could also be safely defined for modules that were implemented in Python, thereby confirming that they could be considered for real-time applications.

## 1. Introduction

Data that are derived from sensors can be divided into two broad categories according to different technologies: acquisition and real-time usage. Data acquisition can require the management of large data throughput when a large number of sensors or high sampling speeds are involved, but in this case, the latency of the system was not an issue since data were not required to close a feedback loop. Conversely, when data from sensors are used for control, the latency of the underlying systems can heavily affect the stability of the controlled system itself. In this case, real-time systems were involved, for which the correct management of data flows (from sensor to controllers and actuators) had to be provided and the time that was required for operations could not exceed given system-dependent limits. Figure 1 shows a block diagram for a typical real-time system that performs feedback control, acquires signals from sensors (local or remote) and computes the outputs to be sent to actuators (local or remote).

Real-time systems are typically used to drive physical systems, the maximum allowed reaction times of which are dictated by the nature of the involved phenomena. For slow processes, such as large plants, reaction times can be in the order of seconds, while faster processes, such as the power regulation of electrical engines, may require much faster reaction times that are in the order of milliseconds or even less. The minimum reaction times that can be reliably obtained in computer-based real-time systems are in the order of tens of microseconds. Shorter reaction times can be achieved using field programmable gate array (FPGA) components. However, programming, debugging and integrating FPGA components is a very complex task in computer software development and these components are only used when strictly required. The first step in the configuration of real-time systems is the appropriate tuning of the operating system (OS). For this purpose, it is important to guarantee that the reaction time of the OS (for example, after the activation of a user code upon the occurrence of an interrupt) is as short and deterministic as possible. This requires that most, if not all, of the involved resources are exclusively dedicated to real-time tasks. First of all, no threads or processes should be competing for the same processor. On a general purpose OS, such as Linux, this is normally achieved by insulating a subset of the available cores and assigning the insulated cores to the tasks that are involved in real-time operations. Another issue is interrupt dispatching, which can appropriate computing time from a given core even when it is exclusively assigned to a given thread. In Linux, it is possible to control interrupt dispatching to a certain extent (the dispatching of some interrupts, such as timer interrupts, is normally not controllable), thus reducing the jitters in reaction times. Finally, sharing other resources, such as buses and network interfaces, may still cause jitters in reaction times, even though these typically happen on a reduced scale in respect of core contention, unless very high data throughput is involved. The duplication of network interfaces can be adopted for distributed real-time systems in order to develop two networks: one that is dedicated to the applications and another that is dedicated to Linux management.

As well as properly tuning the operating systems, the careful programming of the involved software modules is required to guarantee deterministic execution times. For this purpose, it is important to remove all possible sources of non-determinism in execution times from the code. This implies avoiding iterative solutions that are based on convergence criteria in favor of algorithms with a fixed maximum number of cycles. Moreover, memory usage must be carefully planned to avoid dynamic memory allocation during real-time computation while pre-allocating all of the required structures during system initialization. For similar reasons, recursive implementations should be avoided as they can cause uncontrolled stack allocation.

The software components that are involved in real-time systems can be divided in two main categories:*Data flow management*. This is the code that is involved in the movement of signals among the system components. Data originate from sensors (e.g., analog to digital converters), network interfaces and other software modules (e.g., the pre-elaboration of sensor data), then the signals are consumed by actuators, network interfaces and other software modules that perform the required real-time computations. For peripherals, software drivers are required and generally, a given software layer supervises all communication both inside (i.e., memory based) and outside (i.e., hardware interfaces and network communication) of the involved computers.*Computation management*. This is the implementation of the required feedback algorithms. For non-trivial systems, several computation modules are involved, which perform the pre- and post-processing of the data in addition to controlling the computations.

In real-world systems, data flow management is normally provided by dedicated frameworks. The use of software frameworks is preferable to novel implementations for various reasons, including the following:It speeds up system implementation because I/O and communication functions are carried out by ready components, in most cases, which only require proper configuration. This also holds for computation components when a component is already available for a given computation block. In most cases, when a component is not available, the framework allows the integration of a new one, which is developed once and then reused when required.It improves the quality of the deployed system compared to that of a system that was developed from scratch because reused components have likely already been used and tested in other systems.

C++ is commonly adopted in real-time systems as it simultaneously offers the possibility for high-level programming, thanks to its object oriented features, together with object constructors and destructors to organize memory usage. This particularly holds for data flow management and has also been used for real-time computations. C++ is currently the most effective performance solution and, therefore, the vast majority of newly developed real-time applications are written in C++. However, C++ expertise is less common compared to other programming languages. Indeed, outside of real-time applications, Python is quickly replacing other languages in many applications thanks to the possibility for rapid code development and the availability of a variety of Python modules that cover most application fields.

Since it is an interpreted language, there is no explicit way of evaluating how internal data structures are handled in Python and, more importantly, when memory allocation is performed, which could potentially introduce unexpected delays in computation times due to the activity of garbage collectors. However, provided that the performance of Python modules is still acceptable for the requirements of a given system, the integration of Python modules in real-time frameworks could provide a significant contribution to the development of real-time systems due to the fact that Python knowledge is considerably more widespread.

For the above reasons, this study aimed to quantify the penalties in computation times and jitters that were incurred when Python components were introduced into real-time systems. For this purpose, it was first necessary to define the field of application, then the reference OS and the framework for data management and, finally, an application example, within which the performance of C++ and Python components could be compared using the same OS optimization and underlying framework configuration.

The chosen field of application was magnetic confinement fusion, which is gaining increasing amounts of interest as a source of energy. For this reason, a growing number of laboratories are carrying out active research on the magnetic confinement of the high-temperature ionized gas (plasma) in which fusion reactions occur [1]. A common requirement of these experiments is a real-time control system that can hold the high-temperature plasma in a container by means of electromagnetic fields, thereby avoiding any contact between the plasma and the container walls, which would lead to the immediate damage of the container itself [2,3,4,5,6,7,8,9,10,11,12,13,14,15,16]. Typical reaction times for real-time systems are in the order of milliseconds, but faster phenomena, such as vertical instabilities, require reaction times that are in the order of tens of microseconds and reach the limits of computer-based real-time systems. Real-time plasma control requires the integration of several software components that derive the plasma parameters from sets of sensor signals [17,18], which are generated by plasma diagnostic systems, in order to compute sets of outputs for system actuators, which are typically represented by the power supplies for the coils that generate the required electromagnetic fields. The derivation of the plasma parameters typically involves rather complicated computations that are based on physical models of plasma behavior [19,20,21,22], which are more and more frequently developed using Python programming.

In the context of fusion research, MARTe2 is a widely used framework for real-time plasma control [23,24,25,26,27]; therefore, it was considered here for the implementation of the test cases. Since MARTe2 was written in C++, the integration of Python modules required the development of generic C++ wrappers for Python code. Finally, two sample applications were considered. In the first example, a PID computation was implemented in C++ and Python, then its execution times and jitters were compared. The second example considered sensor data acquisition from an ADC module and real-time FFT execution in order to derive the main frequencies of the input signals.

The rest of this paper is organized as follows. Section 2 describes the real-time framework that was used to implement the test cases and Section 3 shows how the test cases were implemented within the framework. Section 3 describes how the Python modules were integrated into the system. Section 4 discusses the performance measurements of the C++ and Python modules. Finally, Section 5 presents our final remarks about the applicability of these components in real-time systems.

## 2. The Real-Time Framework

Real-time systems implement different functions, which can be classified as follows:*Input/output (I/O) management*. This is the interaction between the hardware devices that implement the system inputs (sensors) and produce the system outputs (actuators). For example, in fusion experiments, the inputs are electromagnetic sensors that discern voltages, currents and magnetic fields [17,18], which are acquired via ADC devices. The actuators are the power supplies for the coils that are used to generate the electromagnetic fields that are required to shape the plasma and, in particular, avoid any contact between the plasma and the container walls. The references to the power supplies are either analog, i.e., produced by a digital to analog converter (DAC), or sent directly via digital links between control systems and power supply controllers.*Communication*. This carries out the required data flows in real time among the involved components. Depending on the complexity of the real-time system, data communication is either limited to memory communication when systems are hosted on single computers or requires network communication when systems are distributed.*Algorithm computation*. This computes the actuator references based on the sensor inputs and current system states.

While the functions of the last class generally depend on the nature of the system, components that belong to the first two classes can be reused in different systems by simply using a different configuration. For example, the support code for a given ADC device could be reused in all systems in which that device is used. Similarly, software components that carry out network communication using a given protocol can be reused in different distributed systems using a different topology. It is worth noting that in some cases, even computation components can be reused, especially when they carry out common computations, such as efficient matrix multiplication or Fourier analysis. For this reason, it makes sense to reuse as many system components as possible within software frameworks.

Here, the MARTe2 real-time framework was used. MARTe2 provides a set of components that implement I/O, communication and computation functions, which were written in C++ with high-quality standards (which are required for certain applications, such as fusion experiments). These components are defined and connected within a framework and are configurable via a textual description. New components can be developed and integrated into MARTe2 for specific applications. In particular, the framework defines two classes of components, as follows:I/O components (also called *data sources*), which implement hardware and communication interfaces;Computation components (also called generic application modules or *GAM*s), which implement the units of computation. Every GAM defines a set of inputs and outputs that can be connected to other components (data sources or GAMs) within the system configuration.

New data sources and GAMs can be implemented as C++ classes that inherit generic functionality from sets of superclasses, which are provided by the framework.

## 3. Test Case Descriptions

The first test case performed PID computation and we compared the execution times of the C++ and Python GAM implementations. The MARTe2 components that were used in this case were:A GAM for PID computation, which was already available in the MARTe2 toolkit;A Python wrapper GAM that was used to wrap a custom Python PID module, as described in Section 3 (the same wrapper GAM was also used in the second test case with a different target Python program).

The second test case consisted of the ADC sampling and FFT computation of the acquired samples, which were performed over sets of 2048 samples. The following MARTe2 components were used for this purpose (Figure 2):A data source for the ADC, which acquired a given input signal at a sampling rate of 2 MHz and delivered the acquired data in blocks of 2048 samples;A GAM that implemented the fast Fourier transform (FFT) of its inputs and was connected with the output of the data source (these two components cyclically exchanged arrays of 2048 float samples) and the overall real-time loop had a period of 1.024 ms as the cycle was synchronized by a 2 MHz ADC sampling clock;A Python wrapper GAM to wrap the Python FFT computation, which was carried out using Numpy, as described in Section 3 (this was the same wrapper GAM that was developed for the PID test case).

While the first component was already available in the MARTe2 toolkit, the GAM for the FFT computation was specifically developed for the test case using FFTW [28], which is a library of highly efficient FFT computations that is widely used within the scientific community.

MARTe2 offers a set of performance measurement tools that do not perturb real-time execution, which were used to derive the performance results that are presented in Section 4. In the second test case, it was assumed that no samples were lost when the overall cycle execution time did not exceed 1.024 ms. This time was the sum of the time that was required by the ADC data source to read the samples from the ADC devices, the time that was required to transfer the set of 2048 samples to the memory and the time that was required to perform the FFT computation. The first two times turned out to be very small compared to the FFT computation times; therefore, we concentrated on the latter. In order to state the proper real-time behavior, i.e., ensure that no samples were lost in the cycles, it was important not to consider average computation alone, but also its jitters. In other words, we were interested in the longest execution time: when the worst-case execution time (WCET) was less than the cycle time (also considering a safety margin), then we could ensure that the system behaved as expected, i.e., no samples were lost. Note that jitters can be less predictable in more complex systems that involve interacting control loops with different cycle frequencies.

### Python Integration

As stated previously, new components were integrated into MARTe2 by implementing a new C++ class. Since writing wrappers to integrate Python into MARTe2 GAMs is not trivial, writing different wrappers for every different Python function to be exported to MARTe2 is not desirable. For this reason, in this work, we developed a generic wrapper for MARTe2 that could be used with any Python module, regardless of the numbers, types or dimensions of its inputs, outputs and parameters. Every MARTe2 component could check the layout of its inputs, outputs and parameters, as defined in the specific framework configuration, in order to check whether the current configuration was compatible with the module interface. This check was performed during system initialization, before entering the real-time loop. In order to let a single Python wrapper be reused for a variety of Python modules (each with its own specific set of inputs, outputs and parameters), the wrapper first needed to look at the layout of the specific Python interface using the Python introspection functions and then check that interface against the current MARTe2 configuration using the MARTe2 introspection functions. When the two interfaces were compatible (i.e., when the numbers, types and dimensions of the inputs, outputs and parameters that were declared in Python matched the current component configuration in MARTe2), the check was passed and MARTe2 could enter the real-time loop. The specific implementation of the wrapper copied the GAM inputs from MARTe2 into the Python memory space, called the target Python module step routine and then copied the outputs back from the Python memory to MARTe2.

In this way, new Python functions were readily integrated into MARTe2 using a single GAM that required the name of the target Python module as a mandatory parameter. It is worth noting that MARTe2 offers the possibility of performing extensive quality tests via a set of user-provided unit tests, which also produce an automated analysis of code coverage. Unit tests can also be defined for integrated Python modules; however, the automated code coverage analysis is not possible for these modules.

In order to provide a meaningful comparison and reduce performance differences that were due to different OSs or framework behavior as far as possible, the same optimization process was carried out for the OS configurations in both cases by segregating the cores and excluding them from interrupt dispatching and disabling processor idling. Moreover, the same configuration for the MARTe2 framework was used in both cases, only the GAM that carried out the PID or FFT calculations was changed (C++ or Python).

## 4. Performance Measurements

In the first test case, the execution time of the PID computation that was performed cyclically at 1 kHz with a single segregated core assigned to the computation thread was measured for a native C++ PID implementation and a Python implementation. The execution time distribution is reported in Figure 3a. As expected, the average execution time and jitters were higher in the Python application than the C++ application. However, the WCET for the Python PID could be safely assumed to be 100 μs; therefore, this module could be used whenever that WCET was defined.

In the second test case, FFT computations were performed over sets of samples that were acquired from an ADC device at 2 MHz.

As stated previously, the real-time loop timing was dictated by the 2 MHz ADC sampling clock, but the overall cycle execution time could not exceed 1.024 ms so as not to lose samples.

The loop computation time was dominated by the time that was required to perform the FFT computation, which was recorded for every loop using the performance collection tools of MARTe2. Figure 3b shows the computation time histograms for the C++ and Python implementations. It can be seen that the difference in average execution times between the two different implementations was less than 100 μs. More importantly, the jitters in the execution times turned out to be very similar, which led to the conclusion that the Python module could well be used in the considered configuration as the margin in the computation time (i.e., the difference between the cycle time and the maximum execution time) was large enough to guarantee safe real-time execution.

It is worth noticing that in the FFT case, the jitters in the execution time of the Python module were very similar to those of the C++ version, unlike the PID test case. The reason for this was in the second test case, the Python code basically acted as a wrapper for the Numpy functions, which in turn used compiled code [29].

In order to increase the determinism in the execution times, garbage collection was excluded in both cases, which ensured that the overall object reference count did not increase over time. Particular care was required for the C++ code wrapper because, unlike Python code, every Python object that was created by the wrapper has to be properly dereferenced. When using Python for real-time applications, circular object dependencies should be totally avoided because they prevent the reclamation of memory space when the object reference count drops to zero, unless a garbage collector is activated when the overall allocation count exceeds a given threshold. Garbage collection has an impact on real-time system performance because its occurrence and execution is not predictable. The effects of garbage collectors are clearly shown in Figure 4 and Figure 5, in which a Python object was intentionally not dereferenced in the Python wrapper during real-time execution. The spikes in the time evolution (Figure 4) and the distribution of the execution times (Figure 5) reflect the periodic activation of the garbage collectors whenever the reference count (which was intentionally increased over time) exceeded the garbage collector threshold.

## 5. Conclusions

The presented test cases compared Python and C++ implementations for real-time components that performed PID and FFT computations, respectively. The components were tested using the same underlying Linux optimization and the same software framework in order to highlight differences in execution times and jitters that were exclusively due to the different languages being used.

The outcome of these test cases was that Python could also be considered as a candidate for the elaboration of data from sensors in real-time control systems, thereby disproving initial concerns about its usability in real time. In particular, the use of Python modules allowed us to take advantage of the powerful Numpy library of scientific computations. However, careful coding was also required in Python in order to avoid potential sources of delays and jitters in execution times. In general, the need for garbage collectors could be removed by avoiding circular dependencies in memory objects.

## Figures and Tables

**Figure 1 sensors-22-06847-f001:**
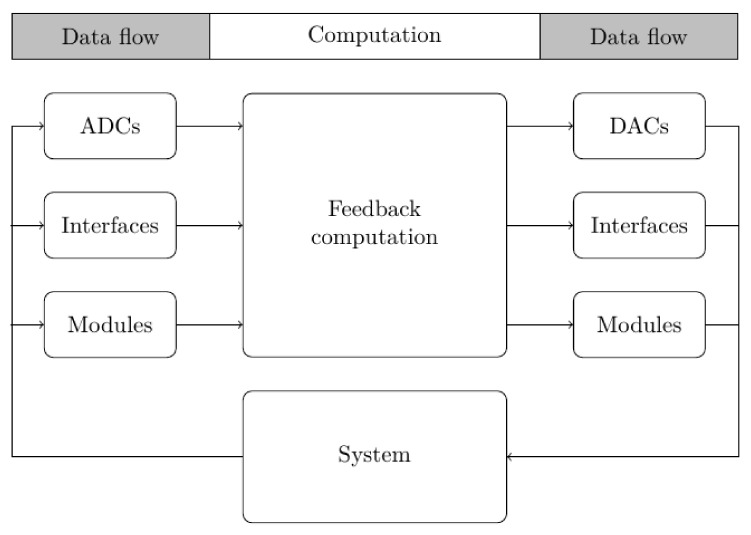
An outline of the software components in a typical real-time system that performs feedback control. Input data come from analog to digital (ADC) converters that read sensor inputs, network interfaces (when the system is distributed among different computers in a local area network (LAN)) or other software modules that perform input data pre-elaboration. Output data are directly sent to digital to analog (DAC) converters in order to drive actuators, network interfaces (when the actuators are on a different system) or other software modules for post-processing elaboration.

**Figure 2 sensors-22-06847-f002:**
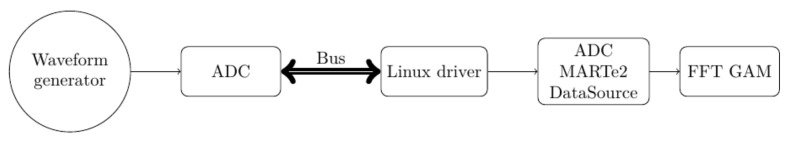
A block diagram of the second test case: the input waveform was acquired by an ADC module that sent data samples over a communication bus (PCIe) to a computer; data were then made available to the ADC data source software component via the Linux driver for the ADC; finally, data were collected by the ADC data source and grouped into blocks of 2048 samples, which were sent to the GAM that was performing the FFT computation.

**Figure 3 sensors-22-06847-f003:**
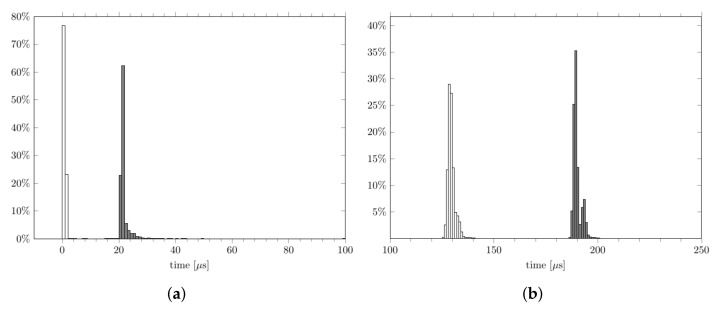
Cycle times: (**a**) the PID computation for the C++ (white) and Python (gray) applications without garbage collectors; (**b**) the real-time FFT computations for the C++ (white) and Python (gray) applications without garbage collectors.

**Figure 4 sensors-22-06847-f004:**
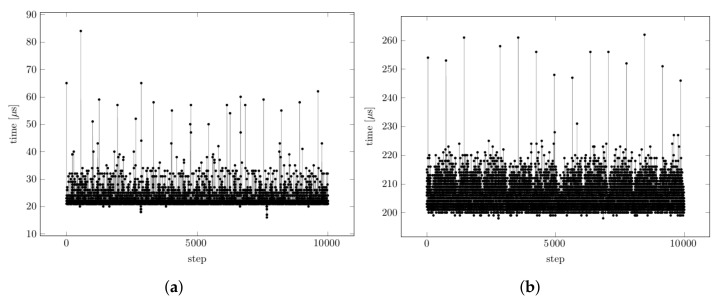
The evolution of the Python module execution time with interference from the garbage collectors, which resulted in regular spikes in the cycle times: (**a**) PID computation; (**b**) FFT computation.

**Figure 5 sensors-22-06847-f005:**
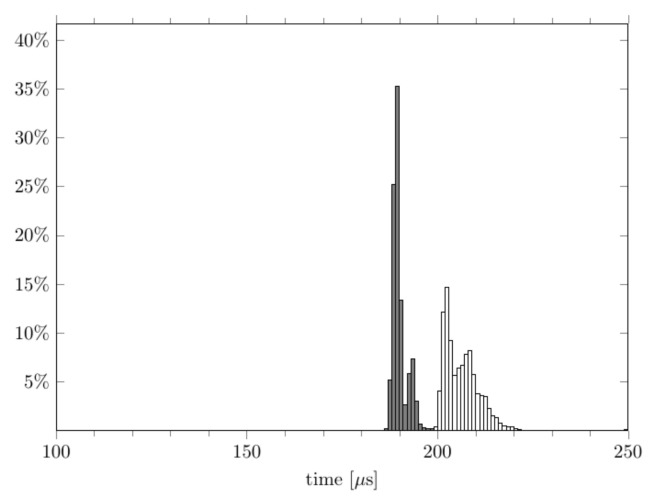
A cycle time histogram for the Python FFT module, both with (white) and without (gray) the action of the garbage collector.

## Data Availability

The MARTe2 Framework is open source and available at the git repository: https://vcis-gitlab.f4e.europa.eu/aneto/MARTe2 (accessed on 6 September 2022).

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
