# Peer review of "Using Python Modules in Real-Time Plasma Systems for Fusion"

_sensors, 2022, doi:10.3390/s22186847_

Round 1
Reviewer 1 Report
The paper is well written and clear.
The methodology is sound.
The topic is quite specific and circumscribed around a given community, but the outcome might turn out to be useful also in other technological fields and to other communities.
The introduction is quite broad and could be a bit longwinded for readers that are already familiar with the discussed topic. However, I acknowledge that it has some didactical value for people that are not expert in the field.
From the viewpoint of the technical content, I do not have any major remark.
Some minor comments:
- If there is no other specific reason, I would consider to write "Python" instead of "python".
- The acronym "PID" should be included in the list of abbreviations.
- I would not use italics for measurement units (i.e., write "ms" instead of "ms").
- In page 6, replace "100ms" with "100 ms".
- Please, check the order of the references. They are not listed in the same order as they appear in the manuscript.
Author Response
The authors deeply thank the reviewers and the editors for pointing out the following points in a very constructive way. Changes to the manuscript are described below.
Point 1: If there is no other specific reason, I would consider to write "Python" instead of "python".
Response 1: The word is now capitalized.
Point 2: The acronym "PID" should be included in the list of abbreviations.
Response 2: The acronym has been added to the list of abbreviations.
Point 3: I would not use italics for measurement units (i.e., write "ms" instead of "ms").
Response 3: Measurement units are all plain text now.
Point 4: In page 6, replace "100ms" with "100 ms".
Response 4: A space has been added.
Point 5: Please, check the order of the references. They are not listed in the same order as they appear in the manuscript.
Response 5: Reference order has been adjusted accordingly.
Reviewer 2 Report
The paper describe an interesting comparison and application of Python generic application modules for real time applications. Notwithstanding in the specific case I think it is difficult to undermine C++ in such critical applications, the results are promising and interesting for those who would like to implement feedback control with sensors using a Python framework.
Are computers necessary or also a controller on FPGA for example can be effectively used? Please comment this point in the introduction
Does C++ has any drawback in the considered application or not? This consideration allows the reader to give the correct perspective in understanding the aim of the paper. Also clarify if Python has a future is such applications (and why)
I understand that the paper discusses issues that are really specific to plasma confinement, however it seems strange that the only references that are not from authors of RFX are [7] and [8].
MARTe2 is briefly introduced twice (p. 3 line 123 and p. 4 line 165)
To give a context for the occasional reader, please better clarify, also with a diagram, the structure of the system (from the plasma to the computational modules and the return of the feedback.
Generally speaking my impression is that the paper would require more visual support and details of the system for the reader that is not expert in the field. This is particularly critical because the paper is not submitted to a fusion engineering journal
Are the quality standards of the Python implementation similar to the C++ ones? Is the software ready to be run in the application?
Use the word "Python" always capitalised, as used on the python.org website
Author Response
Response to Reviewer 1 Comments
The authors deeply thank the reviewers and the editors for pointing out the following points in a very constructive way. The manuscript has indeed significantly improved thanks to the really helpful comments. Changes to the manuscript are described below.
Point 1: Are computers necessary or also a controller on FPGA for example can be effectively used? Please comment this point in the introduction
Response 1:
We have added in the introduction the following couple of sentences for introducing FPGAs:
"Shorter reaction times can be achieved with Field Programmable Gate Arrays (FPGAs) components. However, programming, debugging and integrating FPGA components is a much more complex task in respect of computer software development, and such components are only used where strictly required."
Point 2: Does C++ has any drawback in the considered application or not? This consideration allows the reader to give the correct perspective in understanding the aim of the paper. Also clarify if Python has a future is such applications (and why)
Response 2:
C++ always provides better performance, and this has been stressed by the added sentence in the introduction:
“C++ currently represents the most effective solution regarding performance and therefore the great majority of the newly developed real-time applications are written in C++. “
We have better clarified why we are interested here in Python adding the following sentence:
“However, C++ expertise is less common when compared to other programming languages. “
Indeed in the introduction we say that we are going to analyze the penalty in respect of C++ when using Python modules. So it should be now clearer that C++ is always the best solution regarding performance, and the reason for Python analysis presented here is due to the fact that there are much more Python programmers that C++ ones.
Point 3: I understand that the paper discusses issues that are really specific to plasma confinement, however it seems strange that the only references that are not from authors of RFX are [7] and [8].
Response 3: Additional references were added to better reflect the state of the art of the field.
Point 4: MARTe2 is briefly introduced twice (p. 3 line 123 and p. 4 line 165)
Response 4: The second introduction has been removed.
Point 5: To give a context for the occasional reader, please better clarify, also with a diagram, the structure of the system (from the plasma to the computational modules and the return of the feedback.
Response 5:
We have added a new figure (fig. 1) presenting the generic diagram for a real-time feedback system, describing the components in the figure caption.
Point 6: Generally speaking my impression is that the paper would require more visual support and details of the system for the reader that is not expert in the field. This is particularly critical because the paper is not submitted to a fusion engineering journal
Response 6:
The added figure should make the structure of a typical feedback real-time system (for fusion and also other applications) clearer also for a non expert reader. In order to better clarify the second, more complex, presented application, a schema has been also added (Fig. 2).
Point 7: Are the quality standards of the Python implementation similar to the C++ ones? Is the software ready to be run in the application?
Response 7:
We have added the following sentence, clarifying that most of the quality standard checks performed by the framework can be retained also for Python modules, except code coverage analysis.
“It is worth noting that the MARTe2 gives the possibility of performing extensive quality tests via a set of user provided Unit Tests, providing also automated analysis of code coverage. Units tests can also be defined when integrating Python modules, however automated code coverage analysis is not possible for such modules. “
Point 8: Use the word "Python" always capitalised, as used on the python.org website
Response 8: The word has been capitalised.